# Globally Convergent Policy Search for Output Estimation

**Jack Umenberger**
CSAIL, MIT
umnbrgr@mit.edu

**Max Simchowitz**
CSAIL, MIT
msimchow@mit.edu

**Juan C. Perdomo**
EECS, UC Berkeley
jcperdomo@berkeley.edu

**Kaiqing Zhang**
CSAIL & LIDS, MIT
kaiqing@mit.edu

**Russ Tedrake**
CSAIL, MIT
russt@mit.edu

## Abstract

We introduce the first direct policy search algorithm which provably converges to the globally optimal *dynamic* filter for the classical problem of predicting the outputs of a linear dynamical system, given noisy, partial observations. Despite the ubiquity of partial observability in practice, theoretical guarantees for direct policy search algorithms, one of the backbones of modern reinforcement learning, have proven difficult to achieve. This is primarily due to the degeneracies which arise when optimizing over filters that maintain an internal state.

In this paper, we provide a new perspective on this challenging problem based on the notion of *informativity*, which intuitively requires that all components of a filter's internal state are representative of the true state of the underlying dynamical system. We show that informativity overcomes the aforementioned degeneracy. Specifically, we propose a *regularizer* which explicitly enforces informativity, and establish that gradient descent on this regularized objective – combined with a "reconditioning step" – converges to the globally optimal cost at a $\mathcal{O}(1/T)$ rate.

## 1 Introduction

Data used for prediction and control of real world dynamical systems is almost always noisy and incomplete (partially observed). Sensors and other measurement procedures inevitably introduce errors into the datasets, so designing reliable learning algorithms for these noisy or partially observed domains requires confronting fundamental questions of disturbance filtering and state estimation. Despite the ubiquity of partial observation in practice, these concerns are often underexplored in modern analyses of learning for control that assume perfect observations of the underlying dynamics.

In this work, we study the output estimation (OE) problem or learning to predict in partially observed linear dynamical systems. The output estimation problem is one of the most fundamental problems in theoretical statistics and learning theory. Both in theory and in practice, advances in predicting partially observed linear systems have led to successes in a variety of areas from controls to biology and economics, (c.f. e.g. Athans [1974], Lillacci and Khammash [2010], Gautier and Poignet [2001]). We revisit this classical problem from a modern optimization perspective, and study the possibility of learning the optimal predictor via model-free procedures and direct policy search.

Relative to model-based procedures, which first estimate the underlying dynamics and then return a policy by solving an optimization problem using the estimated model, model-free methods offer several potential advantages. For instance, direct policy search allows one to easily specify the complexity of the policy class over which one searches. In addition, model-free policy search optimizes

36th Conference on Neural Information Processing Systems (NeurIPS 2022).

for performance directly on the true system, rather than an approximate model. As such, there is no gap between the model used for synthesis and the system on which the policy is deployed.

In light of these advantages, there has recently been significant interest from both theoreticians and practitioners in understanding the foundations of model-free decision making. So far, this attention has focused mostly on problems with *full-state* observation such as the linear quadratic regulator (LQR) or fully-observed Markov Decision Processes (MDPs), for which optimal policies are *static* (without memory). Progress on partially observed problems has been complicated by the difficulties associated with optimizing over *dynamic* policies (with memory) that maintain internal state to summarize past observations. In this paper, we provide the first policy search algorithm which provably converges to the globally optimal filter for the OE problem, and shed new light on the intricacies of the underlying optimization landscape.

**The Output Estimation problem.** We study one of the simplest and most basic problems with partial observability: the *output estimation* (OE) problem. In brief, the goal is to search for a predictor of the output $\mathbf{z}(t)$ of a linear dynamical system given partial measurements $\mathbf{y}(t)$. For the *true system* with states $\mathbf{x}(t)$ and dynamics that evolve according to,

$$\frac{\mathrm{d}}{\mathrm{d}t}\mathbf{x}(t) = \mathbf{A}\mathbf{x}(t) + \mathbf{w}(t), \quad \mathbf{y}(t) = \mathbf{C}\mathbf{x}(t) + \mathbf{v}(t), \quad \mathbf{z}(t) = \mathbf{G}\mathbf{x}(t), \quad \mathbf{x}(0) = 0,$$
$$\mathbf{w}(t) \overset{\text{i.i.d}}{\sim} \mathcal{N}(0, \mathbf{W}_1), \quad \mathbf{v}(t) \overset{\text{i.i.d}}{\sim} \mathcal{N}(0, \mathbf{W}_2), \tag{1.1}$$

the goal is to find the parameters $\mathsf{K} = (\mathbf{A}_\mathsf{K}, \mathbf{B}_\mathsf{K}, \mathbf{C}_\mathsf{K})$ of the *filter* (interchangably, *policy*),

$$\frac{\mathrm{d}}{\mathrm{d}t}\hat{\mathbf{x}}(t) = \mathbf{A}_\mathsf{K}\hat{\mathbf{x}}(t) + \mathbf{B}_\mathsf{K}\mathbf{y}(t), \quad \hat{\mathbf{z}}(t) = \mathbf{C}_\mathsf{K}\hat{\mathbf{x}}(t), \tag{1.2}$$

that minimizes the steady-state prediction error,

$$\mathcal{L}_{\text{OE}}(\mathsf{K}) := \lim_{T \to \infty} \frac{1}{T}\left(\mathbb{E}\int_0^T \|\mathbf{z}(t) - \hat{\mathbf{z}}(t)\|^2 \mathrm{d}t\right) = \lim_{t \to \infty} \mathbb{E}\|\mathbf{z}(t) - \hat{\mathbf{z}}(t)\|^2. \tag{1.3}$$

In this paper, we study solving the OE problem via model-free methods, where the goal is to search for the optimal filter parameters $\mathsf{K} = (\mathbf{A}_\mathsf{K}, \mathbf{B}_\mathsf{K}, \mathbf{C}_\mathsf{K})$ using direct policy search without knowledge or estimation of the true system parameters $\mathbf{A}, \mathbf{C}, \mathbf{G}, \mathbf{W}_1, \mathbf{W}_2$; see Section 2 for further details. The OE problem is not a control problem per se, as decisions $\hat{\mathbf{z}}$ do not influence the state evolution of the true system (1.1). Nonetheless, it is an attractive stepping stone for studying model-free control. Like LQG, optimal policies for OE are *dynamic*, i.e. have memory. OE therefore serves as a valuable testbed to develop the theoretical foundations of direct policy search for this important class of policies which has thus far remained poorly understood.

**Related work.** Recent years have witnessed a resurgence of interest in direct policy search for control and reinforcement learning. Fazel et al. [2018] established global convergence of policy gradient methods on discrete-time LQR. Subsequent work has sharpened rates [Malik et al., 2019], analyzed convergence under more general frameworks [Bu et al., 2019], and extended the analysis to continuous-time [Mohammadi et al., 2021]. Beyond LQR, Zhang et al. [2020] analyzed policy search for mixed $\mathcal{H}_2/\mathcal{H}_\infty$ and risk-sensitive control. Sun and Fazel [2021] also considered analysis via convex reformulations, for state feedback problems. For discrete state-action MDPs, Agarwal et al. [2021] established convergence rates for a variety of policy gradient methods, cf. also Bhandari and Russo [2019]. All of these works considered *static* state-feedback policies, with perfect state information. For problems with partial observation – which necessitates the use of *dynamic* policies - the work most relevant to our own is Tang et al. [2021], which studied policy search for LQG. They established that all stationary points corresponding to minimal controllers are globally optimal, and demonstrated that gradient descent may fail to converge to optimal policies.

## 1.1 Contributions

We propose a novel policy search method which provably converges to a globally optimal $\mathcal{L}_{\text{OE}}$ cost. Despite extensive prior work on *static* policy search (see above), our result constitutes the first rigorous guarantee for policy search over *dynamic* policies. While the OE problem admits a convex reformulation [Scherer et al., 1997], it is also well known that its landscape contains suboptimal critical points. This may seem like a contradiction at first, but we demonstrate that such spurious

critical points occur precisely when the policy loses what we term "informativity": that the steady covariance between the controller state and system state, $\boldsymbol{\Sigma}_{12} := \lim_{T\to\infty} \frac{1}{T}[\int_0^T \mathbf{x}(t)\hat{\mathbf{x}}(t)^\top \mathrm{d}t]$, becomes rank-deficient. With this insight, we propose a regularizer $\mathcal{R}$ that ensures the internal state of the learned policy remains "uniformly informative" about the state of the true system, and prove that gradient descent on the *regularized* objective $\mathcal{L}_{\mathtt{OE}}(\cdot) + \lambda\mathcal{R}(\cdot)$ converges at a $\mathcal{O}\left(1/T\right)$ rate to an optimal policy. The efficacy of the approach is illustrated via numerical experiments, cf. Appendix F.6. Code is provided in the supplementary material.

**Our techniques.** Searching over dynamic policies introduces two key challenges: (1) spurious critical points can arise when one or more factors become "degenerate" in a certain way; (2) changes of basis produce a continuum of "equivalent realizations" of the filter K, some of which are poorly conditioned. Similar challenges have been observed in problems with rotational symmetries, e.g. nonconvex matrix factorization. Neither challenge arises when searching over static policies [Fazel et al., 2018]. Our approach is centered around the idea of *convex reformulations* of control synthesis problems, and the following fact regarding functions which admit these reformulations, cf. Appendix H.1 for proof.

**Fact 1.1.** Let $f : \mathbb{R}^{n_x} \to \mathbb{R}$ be a differentiable, possibly nonconvex function such that $\min_{\boldsymbol{x}} f(\boldsymbol{x})$ is finite. Suppose there exists a differentiable function $\Psi : \mathbb{R}^{n_\nu} \to \mathbb{R}^{n_x}$ satisfying the following two properties: (i) the mapping $\Psi$ is surjective, i.e. for all $\boldsymbol{x} \in \mathbb{R}^{n_x}$ there exists $\boldsymbol{\nu} \in \mathbb{R}^{n_\nu}$ such that $\boldsymbol{x} = \Psi(\boldsymbol{\nu})$, (ii) under the change of variables the function $f_{\mathrm{cvx}}(\boldsymbol{\nu}) := f(\Psi(\boldsymbol{\nu}))$ is differentiable and *convex*. Then all first-order stationary points, $\boldsymbol{x}$ s.t $\nabla f(\boldsymbol{x}) = 0$, are globally optimal.

The $\mathtt{OE}$ problem, LQG, and many other related control tasks admit convex reformulations [Scherer et al., 1997, Masubuchi et al., 1998]. Given that gradient descent (under mild regularity assumptions) converges to stationary points, we might hope that Fact 1.1 guarantees that direct policy search on the $\mathtt{OE}$ filter will succeed at finding an optimal policy, when applied to loss functions admitting such convex reformulations. Somewhat surprisingly, we find that this is emphatically not the case: gradient descent on the $\mathcal{L}_{\mathtt{OE}}$ objective fails to reliably converge to optimal solutions (see Section 3.1).[1]

To resolve this paradox, we show that the surjectivity condition of Fact 1.1 may fail for the convex reparametrization of $\mathtt{OE}$: there are filters K with finite cost $\mathcal{L}_{\mathtt{OE}}(\mathsf{K})$, which are not in the image of the reformulation map $\Psi(\cdot)$. We find that degeneracy occurs precisely when *informativity*, defined in Section 1.1 as $\boldsymbol{\Sigma}_{12,\mathsf{K}}$ having full rank, fails to hold. Conversely, when $\boldsymbol{\Sigma}_{12,\mathsf{K}}$ is full-rank, the conditions of Fact 1.1 are met and the parametrization behaves as needed. Thus, we identify *non-informativity* - rank deficiency of $\boldsymbol{\Sigma}_{12,\mathsf{K}}$ - as the fundamental notion of degeneracy corresponding to challenge (1). Motivated by this observation, we introduce a novel "informativity regularizer" $\mathcal{R}_{\mathtt{info}}(\cdot)$ which enforces that $\boldsymbol{\Sigma}_{12,\mathsf{K}}$ is full rank. Our proposed algorithm, IR-PG alternates between gradient updates on the regularized loss $\mathcal{L}_\lambda(\cdot) := \mathcal{L}_{\mathtt{OE}}(\cdot) + \lambda\mathcal{R}_{\mathtt{info}}(\cdot)$, and "reconditioning" steps to ensure well-conditioned realizations of the filters K, thereby addressing challenge (2) above. We stress that our notion of informativity differs from the *minimality* criterion emphasized in Tang et al. [2021], whose limitations we discuss in Section 3.1.

In order to achieve our quantitative converge guarantees, we establish numerous results which may be of independent interest, including: (i) a *quantitative* analysis of the $\mathtt{OE}$ convex reformulation due to Scherer [1995], (ii) novel bounds on the magnitude of solution to Lyapunov equations under the closed-loop $\mathtt{OE}$ filter dynamics. Both arguments appeal to a (quantitative measure of) informativity. Finally, we develop a quantitative analogue of Fact 1.1 via a paradigm we call *differentiable convex liftings*, which allows us to establish a form of gradient dominance that we term weak-PL in reference to the well-known Polyak-Łojasiewicz inequality; cf. Section 4.1 for details.

## 2   Preliminaries

Before presenting our main results in Section 3, we first introduce some relevant definitions, and provide the reader with some relevant background on prediction in partially-observed dynamical systems. We adopt standard notation wherever possible, and for brevity, defer details to Appendix A. As outlined in the introduction, we consider the problem of predicting the outputs of a partially observed linear dynamical system. We refer to the dynamical system defined in Eq. (1.1) as the *true*

---

[1]Failure modes for the $\mathtt{LQG}$ problem were presented by Tang et al. [2021].

*system*, with states $\mathbf{x}(t) \in \mathbb{R}^n$, observations $\mathbf{y}(t) \in \mathbb{R}^m$, and performance outputs $\mathbf{z}(t) \in \mathbb{R}^p$. To ensure the dynamical system has a well-defined steady-state, we assume that $\mathbf{A}$ is stable.

**Assumption 2.1.** The matrix $\mathbf{A}$ is *Hurwitz stable*. That is, the real components of all its eigenvalues are strictly negative: $\Re[\lambda_i(\mathbf{A})] < 0$ for $i \in [n]$.

Because these policies only access the system outputs, and are only evaluated in relation to system outputs, we assume that the true system state is *observable*. Further, we assume that dynamics are subject to sufficiently rich noise excitations.

**Assumption 2.2.** The pair $(\mathbf{A}, \mathbf{C})$ in Eq. (1.1) is observable. That is, the observability Gramian defined as $\mathcal{G}_{\mathrm{obs}} := \int_0^\infty \exp(s\mathbf{A})^\top \mathbf{C}^\top \mathbf{C} \exp(s\mathbf{A}) \mathrm{d}s$ is strictly positive definite.

**Assumption 2.3.** We assume that the noise matrices $\mathbf{W}_1$ and $\mathbf{W}_2$ are strictly positive definite.[2]

As stated previously, we restrict our attention to finding the best dynamic filter within the parametric family described in Eq. (1.2). Note that this family contains the *Bayes optimal predictor* for the $\mathcal{L}_{\mathrm{OE}}$ objective, as we will later describe in more detail. We review a number of basic facts:

**A. Steady state distributions.** We define $\mathcal{K}_{\mathtt{stab}} := \{\mathsf{K} : \mathbf{A}_\mathsf{K} \text{ is Hurwitz-stable}\}$ to be the set of filters such that $\mathbf{A}_\mathsf{K}$ is stable. Under Assumption 2.1, Appendix E.1 shows that this is equivalent to the stability of the closed-loop matrix $\mathbf{A}_{\mathrm{cl},\mathsf{K}}$:

$$\mathbf{A}_{\mathrm{cl},\mathsf{K}} := \begin{bmatrix} \mathbf{A} & 0 \\ \mathbf{B}_\mathsf{K}\mathbf{C} & \mathbf{A}_\mathsf{K} \end{bmatrix}, \quad \mathcal{K}_{\mathtt{stab}} = \{\mathsf{K} : \mathbf{A}_{\mathrm{cl},\mathsf{K}} \text{ is Hurwitz-stable}\}. \tag{2.1}$$

Stability of $\mathbf{A}_{\mathrm{cl},\mathsf{K}}$ is a sufficient condition for $\mathcal{L}_{\mathrm{OE}}(\mathsf{K})$ to be finite, and for the following limiting covariance to be well defined: $\boldsymbol{\Sigma}_\mathsf{K} = \lim_{t \to \infty} \mathbb{E}\begin{bmatrix} \mathbf{x}(t) \\ \hat{\mathbf{x}}_\mathsf{K}(t) \end{bmatrix} \begin{bmatrix} \mathbf{x}(t) \\ \hat{\mathbf{x}}_\mathsf{K}(t) \end{bmatrix}^\top \in \mathbb{S}_+^{2n}$. This steady-state covariance is given by the solution to the continuous-time Lyapunov equation,

$$\mathbf{A}_{\mathrm{cl},\mathsf{K}}\boldsymbol{\Sigma} + \boldsymbol{\Sigma}\mathbf{A}_{\mathrm{cl},\mathsf{K}}^\top + \mathbf{W}_{\mathrm{cl},\mathsf{K}} = 0, \quad \text{where } \mathbf{W}_{\mathrm{cl},\mathsf{K}} := \begin{bmatrix} \mathbf{W}_1 & 0 \\ 0 & \mathbf{B}_\mathsf{K}\mathbf{W}_2\mathbf{B}_\mathsf{K}^\top \end{bmatrix}. \tag{2.2}$$

Notice that $\boldsymbol{\Sigma}_\mathsf{K}$ depends only on $(\mathbf{A}_\mathsf{K}, \mathbf{B}_\mathsf{K})$, but not on $\mathbf{C}_\mathsf{K}$, and that the first $n \times n$ block of $\boldsymbol{\Sigma}_\mathsf{K}$ does not depend on $\mathsf{K}$ at all. To highlight these distinctions, we partition

$$\boldsymbol{\Sigma} = \begin{bmatrix} \boldsymbol{\Sigma}_{11} & \boldsymbol{\Sigma}_{12} \\ \boldsymbol{\Sigma}_{12}^\top & \boldsymbol{\Sigma}_{22} \end{bmatrix}, \quad \boldsymbol{\Sigma}_\mathsf{K} = \begin{bmatrix} \boldsymbol{\Sigma}_{11,\mathrm{sys}} & \boldsymbol{\Sigma}_{12,\mathsf{K}} \\ \boldsymbol{\Sigma}_{12,\mathsf{K}}^\top & \boldsymbol{\Sigma}_{22,\mathsf{K}} \end{bmatrix}, \tag{2.3}$$

and define $\mathcal{K}_{\mathtt{ctrb}} := \{\mathsf{K} \in \mathcal{K}_{\mathtt{stab}} : \boldsymbol{\Sigma}_{22,\mathsf{K}} \succ 0\}$ as the set of filters whose internal state covariance is full rank. We refer to these as the *controllable* policies, as these are precisely the policies for which the pair $(\mathbf{A}_\mathsf{K}, \mathbf{B}_\mathsf{K})$ is controllable.

**B. Equivalent realizations.** There are many different ways of parametrizing a given *dynamic* feedback policy, all of which have exactly the same input-output behavior. Let $\mathbb{GL}(n)$ denote the set of invertible $n \times n$ matrices. In particular, given an invertible matrix $\mathbf{S} \in \mathbb{GL}(n)$ the OE loss of a filter $\mathsf{K}$ is invariant under the following class of similarity transforms:

$$\mathsf{Sim}_\mathbf{S}(\mathsf{K}) : (\mathbf{A}_\mathsf{K}, \mathbf{B}_\mathsf{K}, \mathbf{C}_\mathsf{K}) \mapsto (\mathbf{S}\mathbf{A}_\mathsf{K}\mathbf{S}^{-1}, \mathbf{S}\mathbf{B}_\mathsf{K}, \mathbf{C}_\mathsf{K}\mathbf{S}^{-1}). \tag{2.4}$$

Formally, for any $\mathsf{K} \in \mathcal{K}_{\mathtt{stab}}$ and any $\mathbf{S} \in \mathbb{GL}(n)$, $\mathcal{L}_{\mathrm{OE}}(\mathsf{K}) = \mathcal{L}_{\mathrm{OE}}(\mathsf{Sim}_\mathbf{S}(\mathsf{K}))$. We say that $\mathsf{K}$ and $\mathsf{K}'$ are *equivalent realizations* if they are related by a similarity transformation $\mathsf{Sim}_\mathbf{S}(\mathsf{K}) = \mathsf{K}'$ for some $\mathbf{S} \in \mathbb{GL}(n)$. Note that the set $\mathcal{K}_{\mathtt{ctrb}}$ is also preserved under similarity transformation.

**C. Optimal policies.** The landmark result by Kalman shows that for the system defined by $(\mathbf{A}, \mathbf{C}, \mathbf{W}_1, \mathbf{W}_2)$ the Kalman filter $\mathsf{K}_\star = (\mathbf{A} - \mathbf{L}_\star\mathbf{C}, \mathbf{L}_\star, \mathbf{G})$ achieves minimal $\mathcal{L}_{\mathrm{OE}}$ loss. Here, $\mathbf{L}_\star$ is the Kalman gain which is defined in terms of the solution of the following Riccati equation:

$$\mathbf{A}\mathbf{P}_\star + \mathbf{P}_\star\mathbf{A}^\top - \mathbf{P}_\star\mathbf{C}^\top\mathbf{W}_2^{-1}\mathbf{C}\mathbf{P}_\star + \mathbf{W}_1 = 0, \qquad \mathbf{L}_\star = \mathbf{P}_\star\mathbf{C}^\top\mathbf{W}_2^{-1}. \tag{2.5}$$

We define the set of optimal filters $\mathcal{K}_{\mathtt{opt}}$ to be those which are equivalent to the Kalman filter:

$$\mathcal{K}_{\mathtt{opt}} := \bigcup_{\mathbf{S} \in \mathbb{GL}(n)} \{\mathsf{Sim}_\mathbf{S}(\mathbf{A} - \mathbf{L}_\star\mathbf{C}, \mathbf{L}_\star, \mathbf{G})\}. \tag{2.6}$$

---

[2]We may relax this assumption to $(\mathbf{A}, \mathbf{W}_1)$ being controllable.

**D. Restricted problem setting.** The problem description outlined above, including Assumptions 2.1 to 2.3, constitutes the standard OE problem, well-known in control theory, cf. [Doyle et al., 1989, §IV.D]. In this paper, we will make the following additional assumption that restricts the class of OE problems we consider. Further discussion on the utility and necessity of this assumption (for our analysis) is provided in Section 3.2 and Appendix D; the latter also shows that Assumption 2.4 holds for "generic" problem instances.

**Assumption 2.4.** The (cannonical) optimal policy $(\mathbf{A} - \mathbf{L}_\star\mathbf{C}, \mathbf{L}_\star)$ is controllable, i.e. $\int_0^\infty \exp(s(\mathbf{A} - \mathbf{L}_\star\mathbf{C}))\mathbf{L}_\star\mathbf{L}_\star^\top \exp(s(\mathbf{A} - \mathbf{L}_\star\mathbf{C}))^\top \mathrm{d}s$ is strictly positive definite.

**E. Interaction protocol.** In the spirit of model-free methods, we introduce algorithms which work only assuming access to cost and gradient evaluation oracles. We abstract away the particular implementation of these oracles to simplify our presentation and assume that they are exact, in order to focus on the overall optimization landscape of the OE problem. More formally, for any filter $\mathsf{K} \in \mathcal{K}_{\mathtt{stab}}$, $\mathsf{Eval}(\mathsf{K}, \mathcal{L}_{\mathtt{OE}})$ returns the OE cost, $\mathcal{L}_{\mathtt{OE}}(\mathsf{K})$ and $\mathsf{Grad}(\mathsf{K}, \mathcal{L}_{\mathtt{OE}})$ return the gradient of the OE cost, $\nabla\mathcal{L}_{\mathtt{OE}}(\mathsf{K})$. Despite this simplification, we would like to again emphasize that these can be efficiently approximated in finite samples, and purely on the basis of *observations* $\mathbf{y}_t$ subsampled in discrete intervals. For further discussion, please see Appendix C.

Lastly, in addition to standard cost and gradient evaluations of the $\mathcal{L}_{\mathtt{OE}}$ loss, as part of our algorithm, we further require access to gradient and cost evaluations of smooth functions of the stationary-state covariance. Specifically, if $f : \mathbb{S}_+^{2n} \to \mathbb{R}$ is a function of the covariance matrix $\mathsf{K}$, we assume we can compute $\mathsf{Eval}_{\mathrm{cov}}(\mathsf{K}, f)$ which returns the $f(\mathbf{\Sigma}_\mathsf{K})$ and $\mathsf{Grad}_{\mathrm{cov}}(\mathsf{K}, f)$ which returns $\nabla_\mathsf{K} f(\mathbf{\Sigma}_\mathsf{K})$. In Appendix C.2, we show that these oracles can be implemented without direct state access by "subsampling" multiple observations at different time steps.

## 3 Main Results

In this section, we present the main contributions of our work. After demonstrating that the OE cost function contains stationary points that are not globally optimal, we present *informativity-regularized policy gradient* (IR-PG), a direct policy search algorithm based on a novel regularization strategy to preserve *informativity*, introduced in Section 1.1. We state a formal convergence result showing that IR-PG converges to a globally optimal filter at a $\mathcal{O}(1/T)$ rate. The efficacy of IR-PG is illustrated via numerical experiments in Appendix F.6, cf. Fig. 3 in particular to see how informativity regularization dramatically improves the convergence of policy gradient methods.

### 3.1 Existence of suboptimal stationary points

Perhaps the simplest model-free approach to the OE problem is to run gradient descent on $\mathcal{L}_{\mathtt{OE}}(\cdot)$. Under mild assumptions on the loss function $\mathcal{L}_{\mathtt{OE}}$, gradient descent will converge to a first-order stationary point of $\mathcal{L}_{\mathtt{OE}}$. Unfortunately, despite the existence of a convex reformulation [Scherer et al., 1997] and Fact 1.1, the $\mathcal{L}_{\mathtt{OE}}$ loss function contains suboptimal stationary points:

**Example 3.1.** Consider the OE instance given by $\mathbf{A} = -\mathbf{I}_2$, $\mathbf{C} = \mathbf{I}_2$, $\mathbf{W}_1 = 3 \times \mathbf{I}_2$, $\mathbf{W}_2 = \mathbf{I}_2$, and the filter $\mathsf{K}_{\mathrm{bad}}$ given by $\mathbf{A}_{\mathrm{bad}} = -\varepsilon \times \mathbf{I}_2$, $\varepsilon > 0$, $\mathbf{B}_{\mathrm{bad}} = \mathbf{0}_2$, $\mathbf{C}_{\mathrm{bad}} = \mathbf{0}_2$. $\mathsf{K}_{\mathrm{bad}}$ constitutes a suboptimal stationary point of $\mathcal{L}_{\mathtt{OE}}$ for this OE instance.

The example is similar in spirit to Tang et al. [2021, Theorem 4.1]; details are given in Appendix F.1. Importantly, the cost is invariant under perturbations to any single parameter of the filter and $\mathsf{K}_{\mathrm{bad}}$, being equivalent to the zero-filter, is suboptimal. The same is true for any OE instance: every filter with $\mathbf{B}_{\mathrm{bad}} = \mathbf{0}$, $\mathbf{C}_{\mathrm{bad}} = \mathbf{0}$, and $\mathbf{A}_{\mathrm{bad}}$ being stable is a suboptimal stationary point of $\mathcal{L}_{\mathtt{OE}}$. We again emphasize that these suboptimal stationary points arise when the change of variables in the convex reparametrization "breaks down", i.e., these suboptimal filters are not in the image of the reformulation map $\Psi(\cdot)$ of Fact 1.1.

**The perils of enforcing minimality.** A filter $\mathsf{K}$ is *minimal* if $(\mathbf{A}_\mathsf{K}, \mathbf{B}_\mathsf{K})$ is controllable, and $(\mathbf{A}_\mathsf{K}, \mathbf{C}_\mathsf{K})$ is observable. Example 3.1 is the extreme case of a *non-minimal* filter, since $\mathbf{B}_{\mathrm{bad}} = \mathbf{C}_{\mathrm{bad}} = \mathbf{0}_2$. Conversely, as a special case of LQG, the OE problem inherits the property that all stationary points corresponding to *minimal* filters are globally optimal [Tang et al., 2021, Theorem 4.3]. Therefore, it may be natural to ask: *can a local search algorithm enforce minimality to avoid suboptimal stationary points?* A classical result due to Brockett [1976] suggests not: the set of

minimal $n$-th order single-input-single-output transfer functions (e.g. filters) is the disjoint union of $n+1$ open sets. Thus it is impossible for a continuous path to pass from one of these open sets to another without entering a region corresponding to a non-minimal filter, suggesting that a local search algorithm regularized to ensure minimality at every iteration may never converge to the optimal solution (unless a sufficiently large, and lucky, step allows the iterate to hop over the boundary at non-minimality). See Appendix F.2 for further discussion and supporting numerical experiments.

**Futher challenges.** Given the drawbacks of enforcing minimality, one may wonder whether it is sufficient to enforce controllability or observability alone. In Example 3.1 above - and indeed, for all the examples in Tang et al. [2021] of suboptimal stationary points in the LQG landscape - there is a loss of *both* observability ($\mathbf{C}_{\text{bad}} = \mathbf{0}$) and controllability ($\mathbf{B}_{\text{bad}} = \mathbf{0}$). Unfortunately, suboptimal stationary points can occur at controllable policies and observable policies.

**Example 3.2.** Consider the OE instance given by

$$\mathbf{A} = \begin{bmatrix} -1 & 0 \\ 0 & -1 \end{bmatrix}, \quad \mathbf{C} = \mathbf{I}_2, \quad \mathbf{W}_1 = 3 \times \mathbf{I}_2, \quad \mathbf{W}_2 = \mathbf{I}_2,$$

and the filter $\mathsf{K}_{\text{bad}}$ given by

$$\mathbf{A}_{\text{bad}} = \begin{bmatrix} -2 & 0 \\ \gamma & -\gamma \end{bmatrix}, \quad \mathbf{B}_{\text{bad}} = \begin{bmatrix} 1 & 0 \\ 0 & 0 \end{bmatrix}, \quad \mathbf{C}_{\text{bad}} = \begin{bmatrix} 1 & 0 \\ 0 & 0 \end{bmatrix}, \quad \gamma > 0.$$

The filter $\mathsf{K}_{\text{bad}}$ is a controllable but strictly suboptimal first-order critical point of $\mathcal{L}_{\text{OE}}(\mathsf{K})$.

A formal proof is provided in Proposition F.1, but the intuition is as follows: $\mathsf{K}_{\text{bad}}$ is suboptimal because no information about the second state of the true system ever enters the filter (due to sparsity of $\mathbf{B}_{\text{bad}}$). However, $\mathsf{K}_{\text{bad}}$ is controllable as $\gamma > 0$. Further discussion appears in Appendix F.3, where it is also shown that one can similarly construct examples of filters that are observable (but not controllable) that correspond to strictly suboptimal first-order critical points of $\mathcal{L}_{\text{OE}}(\mathsf{K})$. There we also demonstrate that, for the setup in Example 3.2, the minimum eigenvalue of the Hessian $\nabla^2 \mathcal{L}_{\text{OE}}(\mathsf{K}_{\text{bad}})$ can be made arbitrarily close to zero by taking $\gamma$ in $\mathbf{A}_{\text{bad}}$ to be arbitrarily large. This casts doubt on whether approaches based on saddle-point escape with appropriate random perturbations [Jin et al., 2017] or acceleration [Jin et al., 2018] should be expected to work.

## 3.2 A provably convergent algorithm

The previous discussion puts us in a bind: regularizing to preserve minimality may introduce problems related to path-disconnectedness; yet, neither controllability nor observability alone is sufficient to rule out suboptimal stationary points. We identify a stronger condition called *informativity* which is sufficient. We define the set of *informative* filters as

$$\mathcal{K}_{\text{info}} := \{\mathsf{K} \in \mathcal{K}_{\text{stab}} : \text{rank}(\mathbf{\Sigma}_{12,\mathsf{K}}) = n\}.$$

First, we check that all optimal filters lie in this set, under Assumption 2.4 (proof in Appendix E.3).

**Lemma 3.1.** *Under Assumptions 2.1 to 2.4, $\mathcal{K}_{\text{opt}} \subset \mathcal{K}_{\text{info}} \subset \mathcal{K}_{\text{ctrb}}$, and $\mathcal{K}_{\text{info}}$ is an open set.*

Our key insight is that informativity is also *sufficient* to ensure optimality of stationary points, but does not cause path-connectedness issues as it did for minimality (see Appendix G.3 for the proof).

**Theorem 1.** *Let $\mathsf{K} \in \mathcal{K}_{\text{info}}$; then (i) there is a continuous path lying in $\mathcal{K}_{\text{info}}$ connecting $\mathsf{K}$ to some $\mathsf{K}_\star \in \mathcal{K}_{\text{opt}}$ and (ii) if $\nabla \mathcal{L}_{\text{OE}}(\mathsf{K}) = \mathbf{0}$, then $\mathsf{K} \in \mathcal{K}_{\text{opt}}$.*

Theorem 1 suggests that gradient descent with enforced informativity should converge to optimal filters. It is not, however, implied by the landscape analysis of Tang et al. [2021], which focuses solely on *minimal* stationary points. Still, numerous challenges remain: (1) How can one enforce informativity in a smooth fashion? (2) What quantitative measure of informativity provides quantitative suboptimality guarantees on approximate first-order stationary points? (3) Given that $\mathcal{L}_{\text{OE}}(\mathsf{K})$ need not have compact level sets (see Appendix F.3), how does one ensure that the iterates of policy search do not escape to infinity, or reach regions where the loss of smoothness is arbitrarily poor?

**The explained covariance matrix.** In light of Theorem 1, we design a policy search algorithm which ensures that $\mathbf{\Sigma}_{12,\mathsf{K}}$ remains full-rank throughout the search, but does so in a quantitative fashion. Our central object is the *explained covariance matrix*, which measures how much of the covariance of the steady-state system $\mathbf{x}(t)$ is explained by the internal filter state $\hat{\mathbf{x}}_{\mathsf{K}}(t)$ in the large $t$ limit: $\mathbf{Z}_{\mathsf{K}} := \lim_{t\to\infty} (\mathrm{Cov}[\mathbf{x}(t)] - \mathbb{E}[\mathrm{Cov}[\mathbf{x}(t) \mid \hat{\mathbf{x}}_{\mathsf{K}}(t)]])$. When $\mathsf{K} \in \mathcal{K}_{\texttt{ctrb}}$, $\mathbf{Z}_{\mathsf{K}}$ admits an elegant closed-form expression, which provides an alternative definition of $\mathcal{K}_{\texttt{info}}$:

$$\mathbf{Z}_{\mathsf{K}} = \mathbf{\Sigma}_{12,\mathsf{K}}\mathbf{\Sigma}_{22,\mathsf{K}}^{-1}\mathbf{\Sigma}_{12,\mathsf{K}}^{\top}, \text{ so that } \mathcal{K}_{\texttt{info}} = \{\mathsf{K} : \mathsf{K} \in \mathcal{K}_{\texttt{ctrb}} \text{ and } \mathbf{Z}_{\mathsf{K}} \succ 0\}.$$

Since $\mathbf{Z}_{\mathsf{K}}$ is invariant under similarity transformations, $\mathbf{Z}_{\mathsf{K}}$ can be interpreted as a normalized analogue of $\mathbf{\Sigma}_{12,\mathsf{K}}$. Informally, the quadratic form $v^{\top}\mathbf{Z}_{\mathsf{K}}v$ is a sufficient statistic for how much information $\hat{\mathbf{x}}(t)$ contains about the "$v$-direction" of $\mathbf{x}(t)$; see Appendix E.6 for a precise statement.

**Explained-covariance regularization.** We preserve informativity by ensuring our iterates satisfy $\mathbf{Z}_{\mathsf{K}} \succ 0$. To this end, we run gradient descent on the regularized objective for some $\lambda > 0$:

$$\mathcal{L}_{\lambda}(\mathsf{K}) := \mathcal{L}_{\texttt{OE}}(\mathsf{K}) + \lambda \cdot \mathcal{R}_{\texttt{info}}(\mathsf{K}), \quad \text{where } \mathcal{R}_{\texttt{info}}(\mathsf{K}) := \begin{cases} \mathrm{tr}[\mathbf{Z}_{\mathsf{K}}^{-1}] & \mathsf{K} \in \mathcal{K}_{\texttt{info}} \\ \infty & \text{otherwise.} \end{cases} \quad (3.1)$$

The proposed regularizer has several important properties. First, though $\mathcal{R}_{\texttt{info}}$ is nonconvex in $\mathsf{K}$, both $\mathcal{R}_{\texttt{info}}$ and $\mathcal{L}_{\texttt{OE}}$ can be made convex under the *same* change of variables, in the sense of Fact 1.1, cf. Lemma I.2. In addition, $\mathcal{R}_{\texttt{info}}$ is non-negative, and tends to $\infty$ as $\mathbf{Z}_{\mathsf{K}}$ approaches singularity. Furthermore, $\mathcal{R}_{\texttt{info}}$ is invariant under similarity transformations. Next, many of the essential quantities arising in our analysis can be bounded in terms of $\mathbf{Z}_{\mathsf{K}}^{-1}$, justifying $\mathbf{Z}_{\mathsf{K}}$ is a natural quantitative measure of informativity. Lastly, the set of global-minimizers of $\mathcal{R}_{\texttt{info}}(\cdot)$ is precisely the optimal filters for the $\texttt{OE}$ problem, as per the following lemma (see Appendix E.4 for proof).

**Lemma 3.2** (Existence of maximal $\mathbf{Z}_{\mathsf{K}}$). *Under Assumptions 2.1 to 2.3, there exists a unique $\mathbf{Z}_{\star} \succ 0$ such that $\mathbf{Z}_{\star} = \mathbf{Z}_{\mathsf{K}}$ if and only if $\mathsf{K} \in \mathcal{K}_{\texttt{opt}}$, and $\mathbf{Z}_{\star} \succeq \mathbf{Z}_{\mathsf{K}}$ for all $\mathsf{K} \in \mathcal{K}_{\texttt{ctrb}} \setminus \mathcal{K}_{\texttt{opt}}$. Consequently, $\mathcal{K}_{\texttt{opt}} \in \arg\min_{\mathsf{K} \in \mathcal{K}_{\texttt{ctrb}}} \mathcal{R}_{\texttt{info}}(\mathsf{K})$.*

Lemma 3.2 directly implies that the suboptimality of $\mathcal{L}_{\lambda}(\cdot)$ upper bounds the suboptimality in $\mathcal{L}_{\texttt{OE}}(\cdot)$, so we can minimize $\mathcal{L}_{\lambda}$ as a proxy for minimizing $\mathcal{L}_{\texttt{OE}}$.

**Corollary 3.1.** *For any $\mathsf{K}$, we have $\mathcal{L}_{\texttt{OE}}(\mathsf{K}) - \min_{\mathsf{K}'} \mathcal{L}_{\texttt{OE}}(\mathsf{K}') \leq \mathcal{L}_{\lambda}(\mathsf{K}) - \min_{\mathsf{K}'} \mathcal{L}_{\lambda}(\mathsf{K}')$.*

**Reconditioning.** We introduce an additional normalization step between policy updates to ensure the iterates produced by our algorithm have well-conditioned covariance matrices; this in turn ensures the iterates produced by our algorithm remain in a compact set, and that the smoothness of $\mathcal{L}_{\lambda}$ is uniformly bounded. For any filter $\mathsf{K} \in \mathcal{K}_{\texttt{ctrb}}$ such that $\mathbf{\Sigma}_{22,\mathsf{K}} \succ 0$, the reconditioning operator $\mathsf{recond}(\mathsf{K})$ returns a filter $\mathsf{K}'$ which is equivalent to $\mathsf{K}$, but for which $\mathbf{\Sigma}_{22,\mathsf{K}'} = \mathbf{I}_n$. Formally[3],

$$\mathsf{recond}(\mathsf{K}) := \mathsf{Sim}_{\mathbf{S}}(\mathbf{A}_{\mathsf{K}}, \mathbf{B}_{\mathsf{K}}, \mathbf{C}_{\mathsf{K}}), \text{ where } \mathbf{S} = \mathbf{\Sigma}_{22,\mathsf{K}}^{-1/2}. \quad (3.2)$$

Since $\mathsf{K}$ and $\mathsf{K}' = \mathsf{recond}(\mathsf{K})$ are equivalent realizations, we have $\mathcal{L}_{\texttt{OE}}(\mathsf{K}) = \mathcal{L}_{\texttt{OE}}(\mathsf{K}')$, $\mathcal{R}_{\texttt{info}}(\mathsf{K}) = \mathcal{R}_{\texttt{info}}(\mathsf{K}')$, and thus $\mathcal{L}_{\lambda}(\mathsf{K}) = \mathcal{L}_{\lambda}(\mathsf{K}')$.

**Statement of IR-PG.** We can now describe our algorithm IR-PG, whose pseudocode is displayed in Algorithm 1. IR-PG applies gradient descent on the regularized $\mathcal{L}_{\texttt{OE}}$ objective, with an additional balancing step between gradient updates. Appendix B.1 provides a variant where the step size is chosen by backtracking line-search, which enjoys the same rigorous convergence guarantees. To guarantee convergence to an optimal filter (and finiteness of $\mathcal{L}_{\lambda}$), we need to initialize at a filter such that $\mathbf{Z}_{\mathsf{K}_0} \succ 0$, i.e. $\mathsf{K}_0 \in \mathcal{K}_{\texttt{info}}$. Fortunately, random initializations from a continuous distribution satisfy this condition with probability 1 (see Appendix E.7 for a formal statement and proof). Concerning the choice of $\lambda$, the formal guarantees in Section 3.3 below hold for any $\lambda > 0$. Nonetheless, the choice of $\lambda$ can influence the numerical properties of the loss function, such as smoothness and the size of the gradients; cf. Appendix F.6 for further discussion.

---

[3]Our proposed algorithm also works with an approximate balancing $\widetilde{\mathsf{recond}}(\cdot)$, where $\widetilde{\mathsf{recond}}(\mathsf{K})$ returns a $\mathsf{K}'$ which is equivalent to $\mathsf{K}$, and $\|\mathbf{\Sigma}_{22,\mathsf{K}'} - \mathbf{I}_n\| \leq \varepsilon$ for some tolerance $\varepsilon > 0$ (e.g. $\varepsilon = 1/8$).

---
**Algorithm 1** Informativity-regularized Policy Gradient (IR-PG)
---
1: **Input:** Initial $\mathsf{K}_0 \in \mathcal{K}_{\texttt{info}}$, step size $\eta > 0$, regularization parameter $\lambda > 0$
   `% Define` $\mathcal{L}_\lambda(\mathsf{K}) := \mathcal{L}_{\texttt{OE}}(\cdot) + \lambda \mathrm{tr}[\mathbf{Z}_\mathsf{K}^{-1}]$
2: **for** each iteration $s = 0, 1, 2, \ldots$ **do**
3:    **Recondition** $\widetilde{\mathsf{K}}_t = \mathrm{recond}(\mathsf{K}_t)$, where $\mathrm{recond}(\cdot)$ is defined in Eq. (3.2).
4:    **Compute** $\nabla_t = \nabla \mathcal{L}_\lambda(\widetilde{\mathsf{K}}_t)$.
5:    **Update** $\mathsf{K}_{t+1} \leftarrow \widetilde{\mathsf{K}}_t - \eta \nabla_t$.
---

## 3.3 Formal guarantees

We conclude this section by stating the formal convergence guarantee for IR-PG. Our results depend on natural problem quantities, among which is the minimum singular value of $\mathbf{P}_\star$ (as defined in Eq. (2.5)), which we show is always strictly positive.

**Lemma 3.3.** *Let $\mathbf{P}_\star$ be the solution to the Riccati equation in Eq. (2.5). Then under Assumptions 2.1 to 2.3, $\sigma_\star := \lambda_{\min}(\mathbf{P}_\star)$ is strictly positive. Moreover, $\mathbf{P}_\star = \mathbf{\Sigma}_{11,\text{sys}} - \mathbf{Z}_\star$.*

In other words, $\mathbf{P}_\star$ is the limiting conditional covariance of the true state $\mathbf{x}(t)$ given the policy-state $\hat{\mathbf{x}}_\mathsf{K}(t)$ under (any) optimal policy. Lemma 3.3 states that this covariance is nonsingular, i.e. not even optimal policies contain perfect information about any mode of $\mathbf{x}(t)$. Given an initialization $\mathsf{K}_0$, our convergence rate depends polynomially on the following problem parameters:

$$C_{\texttt{sys}} := \max \left\{ \|\mathbf{A}\|, \|\mathbf{C}\|, \|\mathbf{G}\|, \|\mathbf{W}_2\|, \|\mathbf{W}_2^{-1}\|, \|\mathbf{W}_1^{-1}\|, \|\mathbf{\Sigma}_{11,\text{sys}}\|, \sigma_\star^{-1} \right\}. \qquad (3.3)$$

**Theorem 2.** *Fix $\lambda > 0$, $\mathsf{K}_0 \in \mathcal{K}_{\texttt{info}}$. There are terms $\mathcal{C}_1, \mathcal{C}_2 \geq 1$, which are at most polynomial in $n, m, C_{\texttt{sys}}, \lambda, \lambda^{-1}$ and $\mathcal{L}_\lambda(\mathsf{K}_0)$, such that the iterates of IR-PG with any stepsize $\eta \leq \frac{1}{\mathcal{C}_1}$ satisfy*

$$\mathcal{L}_{\texttt{OE}}(\mathsf{K}_s) - \min_\mathsf{K} \mathcal{L}_{\texttt{OE}}(\mathsf{K}) \leq \mathcal{L}_\lambda(\mathsf{K}_s) - \min_\mathsf{K} \mathcal{L}_\lambda(\mathsf{K}) \leq \frac{\mathcal{C}_2}{\eta} \cdot \frac{1}{s}, \quad \forall s \geq 1.$$

The formal guarantee for back-tracking stepsizes is nearly analogous, and given in Appendix B.1. The oracle complexity of IR-PG is described in Appendix B.2. We sketch the highlights of the proof in the following section, after introducing our `DCL` framework. A rigorous proof overview is deferred to Appendix G.2.

## 4 Analysis Framework

In this section, we sketch our analysis in two halves. In the first, we provide a framework for analyzing gradient descent on objectives which admit convex reformulations. This takes place in two parts: first, via our technique of *differentiable convex liftings* (DCLs) we establish a form of gradient dominance we term weak-PL. Second, we show that weak-PL, together with smoothness (i.e. upper bound on the magnitude of the Hessian) implies convergence of gradient descent at a $O(1/t)$ rate. The `DCL` formalizes the developments in the "our techniques" discussion of Section 1.1.

Next, we instantiate the analysis framework for regularized output estimation loss $\mathcal{L}_\lambda$ defined in the previous section. It suffices to establish both (a) smoothness of $\mathcal{L}_\lambda$ and (b) weak-PL by exhibiting a valid DCL. Interestingly, the upper bounds on both the Hessian magnitude and the weak-PL factor depend on the magnitude of $\mathrm{tr}[\mathbf{Z}_\mathsf{K}^{-1}]$, which is precisely what is controlled by the informativity regularizer in Eq. (3.1).

## 4.1 Differentiable convex liftings (`DCLs`)

This section introduces *differentiable convex liftings* (**DCL**s), a rigorous and flexible framework for operationalizing convex reformulations of nonconvex objectives.

**Preliminaries.** To neatly accommodate optimization over constrained domains, we express functions $f : \mathbb{R}^d \to \bar{\mathbb{R}}$ as taking values in the extended reals $\bar{\mathbb{R}} := \mathbb{R} \cup \{\infty\}$.[4] Given such an $f$, we

---

[4]Because we solely consider minimizations, $\bar{\mathbb{R}}$ does not include $-\infty$

denote its domain $\mathsf{dom}(f) : \{\boldsymbol{x} : f(\boldsymbol{x}) \neq \infty\}$ as the set on which $f$ is finite; we say $f$ is *proper* if $\mathsf{dom}(f) \neq \emptyset$; we define its minimal value $\inf(f) = \inf_{\boldsymbol{x} \in \mathbb{R}^d} f(\boldsymbol{x})$. We say $f \in \mathscr{C}^k(\mathcal{K})$ on a domain $\mathcal{K}$ if $f$ is $k$-times continuously differentiable (and finite) on some open set containing $\mathcal{K} \subset \mathbb{R}^d$.

**DCLs**. The DCL is a generic template for convex reformulation that significantly generalizes the setup in Fact 1.1. Rather than relating $f$ directly to a convex $f_{\mathrm{cvx}}$, we "lift" $f$ to a function $f_{\mathrm{lft}}$ by appending auxiliary variables. We then assume a reparametrization $\Phi$ mapping the domain of $f_{\mathrm{lft}}$ to that of $f_{\mathrm{cvx}}$ in this higher dimensional space; intuitively, $\Phi$ is the "inverse" of $\Psi$ in Fact 1.1.

**Definition 4.1.** A triplet of functions $(f_{\mathrm{cvx}}, f_{\mathrm{lft}}, \Phi)$ is a DCL of a proper function $f : \mathbb{R}^d \to \bar{\mathbb{R}}$ if
**(1)** $f_{\mathrm{cvx}} : \mathbb{R}^{d_z} \to \bar{\mathbb{R}}$ is a proper convex function whose minimum is attained by some $\boldsymbol{z}^\star$.
**(2)** For some additional number of parameters $d_\xi \geq 0$, $f_{\mathrm{lft}} : \mathbb{R}^{d + d_\xi} \to \bar{\mathbb{R}}$ is related to $f$ via partial minimization: $f(\boldsymbol{x}) = \min_{\boldsymbol{\xi} \in \mathbb{R}^{d_\xi}} f_{\mathrm{lft}}(\boldsymbol{x}, \boldsymbol{\xi})$.
**(3)** There is an open set $\mathcal{Y} \supseteq \mathsf{dom}(f_{\mathrm{lft}})$ for which $\Phi : \mathcal{Y} \to \mathsf{dom}(f_{\mathrm{cvx}})$ is $\mathscr{C}^1$ and satisfies $f_{\mathrm{lft}}(\cdot) = f_{\mathrm{cvx}}(\Phi(\cdot))$.

Above, $d_z$ and $d_\xi$ are arbitrary dimensions representing the arguments of $f_{\mathrm{cvx}}(\cdot)$ and $f_{\mathrm{lft}}(\boldsymbol{x}, \cdot)$, respectively. The mere existence of a DCL implies that approximate stationary points of $f$ are also approximate minimizers, under conditions elaborated on below:

**Theorem 3.** *Let* $f : \mathbb{R}^d \to \bar{\mathbb{R}}$ *be a proper function with* DCL $(f_{\mathrm{cvx}}, f_{\mathrm{lft}}, \Phi)$. *Then, for any* $\boldsymbol{x} \in \mathsf{dom}(f)$ *at which* $f$ *is differentiable,* $f$ *satisfies the weak-PL condition:*

$$\|\nabla f(\boldsymbol{x})\| \geq \alpha_{\mathrm{DCL}}(\boldsymbol{x}) \cdot (f(\boldsymbol{x}) - \inf(f)), \quad \text{where } \alpha_{\mathrm{DCL}}(\boldsymbol{x}) := \max_{\substack{\boldsymbol{z}^\star \in \arg\min f_{\mathrm{cvx}}(\cdot) \\ \boldsymbol{\xi} \in \arg\min f_{\mathrm{lft}}(\boldsymbol{x}, \cdot)}} \frac{\sigma_{d_z}(\nabla \Phi(\boldsymbol{x}, \boldsymbol{\xi}))}{\|\Phi(\boldsymbol{x}, \boldsymbol{\xi}) - \boldsymbol{z}^\star\|}.$$

Theorem 3 strengthens Fact 1.1 in two respects. For one, it does not impose any smoothness restrictions on $f_{\mathrm{lft}}$ or $f_{\mathrm{cvx}}$; in particular $f_{\mathrm{cvx}}$ can be highly non-smooth and, due to the extended-real function formulation, can also include constraints. And second, the lifting $f_{\mathrm{lft}}$ adds considerable flexibility, which we show is necessary to capture the convex reformulation of OE (Appendix I).

The factor $\alpha_{\mathrm{DCL}}(\boldsymbol{x})$ depends on two quantities. The numerator is the $d_z$-th singular value of $\nabla \Phi(\boldsymbol{x}, \boldsymbol{\xi})$ for any $\boldsymbol{\xi} \in \arg\min f_{\mathrm{lft}}(\boldsymbol{x}, \cdot)$. This captures how large perturbations of $f_{\mathrm{lft}}$'s arguments must be in order to achieve a desired perturbation of the arguments of $f_{\mathrm{cvx}}$, under the reparameterization $\Phi$. The additional arguments in $f_{\mathrm{lft}}$ compared to $f$ adds additional columns to $\nabla \Phi$ thereby making it easier to ensure $\sigma_{d_z}(\nabla \Phi(\cdot)) > 0$. On the other hand, the denominator measures the Euclidean distance between any minimizer of the convex function $f_{\mathrm{cvx}}(\cdot)$ and image of $(\boldsymbol{x}, \boldsymbol{\xi})$ under the reparameterization $\Phi$, and can be bounded under quite benign conditions.

*Proof Sketch.* The formal proof of Theorem 3 (given in Appendix H.2) takes special care to handle cases where $f_{\mathrm{lft}}$ and $f_{\mathrm{cvx}}$ are finite only on restricted domains, as well as possible non-smoothness; still, the main ideas behind are intuitive. For $f_{\mathrm{cvx}}$ being convex, $f_{\mathrm{cvx}}(\boldsymbol{z}) - \inf(f_{\mathrm{cvx}}) = \mathcal{O}(\|\nabla f_{\mathrm{cvx}}(\boldsymbol{z})\|)$. Using the DCL definition and analyzing the inverse image of a point under $\Phi$, we can also establish a gradient dominance result for $f_{\mathrm{lft}}$. Weak-PL for the original function $f$ follows since $f$ is related to $f_{\mathrm{lft}}$ by partial minimization, so its gradients must be larger than those of $f_{\mathrm{lft}}$.

**Gradient descent with DCLs.** We now describe how DCLs yield quantitative convergence guarantees for gradient descent. A more general guarantee accommodating the reconditioning step in IR-PG is deferred to Appendix G.1, and encompasses the bound below as a special case. Given $\alpha > 0$, we say that proper $f : \mathbb{R}^d \to \bar{\mathbb{R}}$ satisfies $\alpha$-*weak-PL* (named after the stronger Polyak-Łojasiewicz condition) on a domain $\mathcal{K} \subset \mathbb{R}^d$ if $f \in \mathscr{C}^1(\mathcal{K})$ and $\|\nabla f(\boldsymbol{x})\| \geq \alpha(f(\boldsymbol{x}) - \inf(f))$. From Theorem 3, we see $f$ satisfies $\alpha_{\mathcal{K}}$-weak PL on $\mathcal{K}$ if $f$ has a DCL and $\alpha_{\mathcal{K}} := \inf_{\boldsymbol{x} \in \mathcal{K}} \alpha_{\mathrm{DCL}}(\boldsymbol{x}) > 0$. To analyze gradient descent, we also require smoothness: we say $f$ is $\beta$-*upper-smooth* on $\mathcal{K}$ if $f \in \mathscr{C}^2(\mathcal{K})$ and for all $\boldsymbol{x} \in \mathcal{K}$, $\nabla^2 f(\boldsymbol{x}) \preceq \beta \mathbf{I}$. The following follows from a standard descent lemma for smooth (though possibly nonconvex) functions.

**Proposition 4.1.** *Let* $\boldsymbol{x}_0 \in \mathsf{dom}(f)$, *and suppose that the level set* $\mathcal{K}(\boldsymbol{x}_0) := \{\boldsymbol{x} : f(\boldsymbol{x}) \leq f(\boldsymbol{x}_0)\}$ *is compact, and* $f$ *satisfies* $\alpha_{\boldsymbol{x}_0}$-*weak PL with* $\alpha_{\boldsymbol{x}_0} > 0$, *and* $\beta_{\boldsymbol{x}_0}$-*upper smoothness on* $\mathcal{K}(\boldsymbol{x})$ *with* $\beta_{\boldsymbol{x}_0} > 0$. *Then,* $\boldsymbol{x}_0$ *lies in the same path-connected component of some minimizer of* $f$, *and for any* $\eta \leq 1/\beta_{\boldsymbol{x}_0}$ *the updates* $\boldsymbol{x}_{k+1} = \boldsymbol{x}_k - \eta \nabla f(\boldsymbol{x}_k)$ *satisfy* $f(\boldsymbol{x}_k) - \inf(f) \leq 2/(k \cdot \alpha_{\boldsymbol{x}_0}^2 \eta)$.

## 4.2 Proof of Theorem 2.

In light of Proposition 4.1 (and its generalization to include reconditioning), it suffices to establish both the weak-PL and smoothness of the loss $\mathcal{L}_\lambda(\cdot)$ on the well-conditioned sublevel set $\mathcal{K}_0 := \{\mathsf{K} : \mathcal{L}_\lambda(\mathsf{K}) \leq \mathcal{L}_\lambda(\mathsf{K}_0), \frac{1}{2}\mathbf{I}_n \preceq \boldsymbol{\Sigma}_{22,\mathsf{K}} \preceq 2\mathbf{I}_n\}$. To establish the weak-PL property, we exhibit a DCL for which $\alpha_{\mathtt{DCL}}(\mathsf{K})$ depends only on the operator norms of $\boldsymbol{\Sigma}_\mathsf{K}, \boldsymbol{\Sigma}_\mathsf{K}^{-1}, \mathbf{Z}_\mathsf{K}$: as well as other system-quantities. Here, $\mathrm{poly}_{\mathrm{op}}(\dots)$ denotes a quantity at most polynomial in the operator norms of its arguments.

**Proposition 4.2.** *For any $\lambda \geq 0$ (non-strict), the objective $\mathcal{L}_\lambda(\mathsf{K})$ admits a DCL $(f_{\mathtt{cvx}}, f_{\mathtt{lft}}, \Phi)$ where the lifted parameter takes the form $(\mathsf{K}, \boldsymbol{\Sigma}_\mathsf{K}) \in \mathcal{K}_{\mathtt{info}} \times \mathbb{S}_{++}^{2n}$, $\mathcal{L}_\lambda(\mathsf{K}) = f_{\mathtt{lft}}(\mathsf{K}, \boldsymbol{\Sigma}_\mathsf{K}) = \min_{\boldsymbol{\Sigma} \in \mathbb{S}_+^{2n}} f_{\mathtt{lft}}(\mathsf{K}, \boldsymbol{\Sigma})$, and where*

$$\sigma_{d_z}(\nabla \Phi(\mathsf{K}, \boldsymbol{\Sigma}_\mathsf{K})) \geq 1/\mathrm{poly}_{\mathrm{op}}\left(\mathbf{A}, \mathbf{C}, \mathbf{W}_2^{-1}, \boldsymbol{\Sigma}_\mathsf{K}, \boldsymbol{\Sigma}_\mathsf{K}^{-1}, \mathbf{Z}_\mathsf{K}^{-1}, \mathcal{L}_{\mathtt{OE}}(\mathsf{K})\right)$$

$$\|\Phi(\mathsf{K}, \boldsymbol{\Sigma}_\mathsf{K})\|_{\ell_2} \leq \left(\max\{n, \sqrt{mn}\} + \sqrt{\mathcal{L}_{\mathtt{OE}}(\mathsf{K})}\right) \cdot \mathrm{poly}_{\mathrm{op}}\left(\mathbf{A}, \mathbf{C}, \mathbf{W}_2^{-1}, \boldsymbol{\Sigma}_\mathsf{K}, \boldsymbol{\Sigma}_\mathsf{K}^{-1}, \mathbf{Z}_\mathsf{K}^{-1}\right).$$

*Furthermore, the norms of the parameters $\mathbf{A}_\mathsf{K}, \mathbf{B}_\mathsf{K}, \mathbf{C}_\mathsf{K}$ satisfy the following bounds:*

$$\max\{\|\mathbf{A}_\mathsf{K}\|_{\mathrm{op}}, \|\mathbf{B}_\mathsf{K}\|_{\mathrm{op}}\} \leq \mathrm{poly}_{\mathrm{op}}\left(\mathbf{A}, \mathbf{C}, \mathbf{W}_2^{-1}, \mathbf{Z}_\mathsf{K}^{-1}, \boldsymbol{\Sigma}_\mathsf{K}, \boldsymbol{\Sigma}_\mathsf{K}^{-1}\right), \quad \|\mathbf{C}_\mathsf{K}\|_{\mathrm{F}} \leq \sqrt{\mathcal{L}_{\mathtt{OE}}(\mathsf{K})/\|\boldsymbol{\Sigma}_\mathsf{K}^{-1}\|}. \tag{4.1}$$

Smoothness is established in (Proposition G.5), which relies on a novel (and rather challenging) bound on the solutions to Lyapunov equations involving $\mathbf{A}_{\mathrm{cl},\mathsf{K}}$:

**Proposition 4.3** (Stability of $\mathbf{A}_{\mathrm{cl},\mathsf{K}}$)**.** *Suppose that $\mathsf{K} \in \mathcal{K}_{\mathtt{info}}$. Then, for any matrix $\mathbf{Y} \in \mathbb{S}^{2n}$, the solution $\boldsymbol{\Sigma}_{\mathsf{K},\mathbf{Y}}$ to the Lyapunov equation $\mathbf{A}_{\mathrm{cl},\mathsf{K}} \boldsymbol{\Sigma}_{\mathsf{K},\mathbf{Y}} + \boldsymbol{\Sigma}_{\mathsf{K},\mathbf{Y}} \mathbf{A}_{\mathrm{cl},\mathsf{K}}^\top + \mathbf{Y} = 0$ satisfies*

$$\|\boldsymbol{\Sigma}_{\mathsf{K},\mathbf{Y}}\|_\circ \leq C_{\mathtt{lyap}}(\mathsf{K}) \cdot \|\mathbf{Y}\|_\circ, \quad \text{where } C_{\mathtt{lyap}}(\mathsf{K}) = \mathrm{poly}_{\mathrm{op}}\left(\boldsymbol{\Sigma}_\mathsf{K}, \boldsymbol{\Sigma}_\mathsf{K}^{-1}, \mathbf{Z}_\mathsf{K}^{-1}, \mathbf{W}_1^{-1}, \mathbf{W}_2^{-1}, \mathbf{C}\right),$$

*and where $\|\cdot\|_\circ$ denotes either the operator, Frobenius, or nuclear norm.*

The regularization $\mathcal{R}_{\mathtt{info}}(\mathsf{K})$ ensures $\|\mathbf{Z}_\mathsf{K}^{-1}\|$ remains bounded on the sublevel $\mathcal{K}_0$; we further show (Lemma G.7) that $\frac{1}{2}\mathbf{I}_n \preceq \boldsymbol{\Sigma}_{22,\mathsf{K}} \preceq \frac{3}{2}\mathbf{I}_n$ implies $\boldsymbol{\Sigma}_\mathsf{K}$ is invertible, and ensures $\|\boldsymbol{\Sigma}_\mathsf{K}\|, \|\boldsymbol{\Sigma}_\mathsf{K}^{-1}\|$ are uniformly bounded. Thus, the weak-PL constant and smoothness parameters are uniformly bounded on $\mathcal{K}_0$, concluding the proof. We stress that the proofs of Proposition 4.2 and Proposition 4.3 require several novel technical arguments, which may be of independent interest. A full proof roadmap, including formal statements of the aforementioned results, is given in Appendix G.2. The fact that $\|\boldsymbol{\Sigma}_\mathsf{K}\|, \|\boldsymbol{\Sigma}_\mathsf{K}^{-1}\|, \|\mathbf{Z}_\mathsf{K}^{-1}\|$ appear throughout the analysis suggests that (a) informativity as measured by $\mathbf{Z}_\mathsf{K}^{-1}$, and (b) the conditioning of $\boldsymbol{\Sigma}_{22,\mathsf{K}}$ may be fundamental to the OE landscape.

**Proof of Theorem 1.** The proof of Theorem 1 is simpler, as it relies mainly on the *existence* of a DCL for which $\alpha_{\mathtt{DCL}}$ does not vanish. See Appendix G.3 for details.

## 5 Conclusion

In this work, we introduce the first direct policy search algorithm which converges to the globally optimal *dynamic* policy for the output estimation problem. We hope that our analysis serves as a valuable starting point to study direct policy search for reinforcement learning and control problems with partial observations, in which the relevant classes of policies are dynamic ones that maintain an internal state. We also hope that both our proposed principle of "informativity", and our technical contributions around convex reformulations, continue to prove useful in future work.

## Acknowledgments and Disclosure of Funding

The authors acknowledge Pablo Parrilo and Tobia Marcucci for helpful discussions. J.U. is supported by NSF Award No. EFMA-1830901, Office of Naval Research (ONR) Award No. N00014-18-1-2210 and ONR Award No. N00014-17-1-2699. M.S. is supported by Amazon.com Services LLC, PO# #D-06310236 and the MIT Quest for Intelligence. K.Z. is supported by a Simons-Berkeley Research Fellowship.

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
