# OpenReview forum: "Globally Convergent Policy Search for Output Estimation"
_NeurIPS.cc/2022/Conference — NeurIPS 2022 Accept_

### Official Review · Reviewer_zdF9 · 2022-07-06

**Rating:** 7
**Confidence:** 1
**Soundness:** 3 good
**Presentation:** 2 fair
**Contribution:** 3 good

**Summary:**

The paper deals with state-space models with linear Gaussian transition and emission models (related terms for this setup are output estimation, Kalman filtering, multivariate autoregressive Gaussian state-space models). This work focuses on continuous-time problems without control inputs. The authors propose a gradient-based algorithm that uses a regularizer term and show that it converges to the global optimum.

**Questions:**

* Can you elaborate on the connection between your work and control problems? I can see that both LQG and OE require a hidden state that is maintained over time, but the same can be said about any state estimation problem and any POMDP. What I'm missing here is a decision-making aspect, where the outputs of the policy would affect which future states are visited.

* If we view OE as a control problem, the "cost" of this problem has a very specific form: $(Gx - u)^2$ (where I replaced $\hat z$ from eq. 1.3 or $C_K \hat x$ from eq. 1.2 with u to highlight the fact that it is a control). Do you see this as an issue in terms of how limiting it is? I don't think I can think of many control problems that have such a cost.

**Limitations:**

I don't have any concern for societal impact. The authors do discuss the precise mathematical perimeter where their results apply.

**Strengths And Weaknesses:**

### Strengths

* The paper identifies a vulnerability in gradient-based optimization of filter matrices and proposes a solution. Both problem and solution are based on a well-motivated mathematical analysis.

* The authors find a new condition that qualifies a critical point as the global optimum. Crucially, the set of points that satisfy the condition is amenable to gradient descent. This would be a landmark result if it is novel (and as far as I can tell it is, but as I've indicated with my confidence score, my expertise is limited).

### Weaknesses

* The paper's connections to control problems are somewhat weak. The authors themselves acknowledge that the output estimation problem is not a control problem. Still, the paper places a heavy emphasis on framing this state estimation problem as a control problem. I am not sure I agree with the usage of terms such as policy gradient and policy search, for instance. None of this disqualifies the paper's contributions in other areas, but if the authors wish to present their work as related to reinforcement learning and optimal control, I believe the connections there need to be explored more in the text.

* I find the structure of the paper a bit difficult to follow. The paper presents many mathematical results before it becomes clear why these are necessary, which makes it harder to follow the author's line of reasoning. No doubt my own lack of familiarity with the content played a role here though.

---

> ### Author Response · Authors · 2022-08-01
> **Initial response to Reviewer zdF9**
>
> Thank you for your feedback, thoughtful critique, and questions.
> We certainly appreciate the concern that output estimation (OE) is not a "control" problem per se,
> and we regret that our initial submission failed to frame the problem correctly.
> While we felt that the OE was an ideal stepping stone to tackle the challenges of partial observation in the study of loss landscapes of actual control problems,
> it is true that the actions taken by the policy (i.e. the predictions made by the filter) do not influence the state evolution of the system; in this sense, OE is not a "control" problem.
> In a broader sense though, OE is a problem of interest to the "control community".
> For instance, it is well understood that linear output estimation (i.e. the filtering problem) is a ``dual'' problem to linear quadratic control.
> Moreover, as we sought to highlight, OE is one of the subproblems solved when constructing solutions to the partially observed `linear quadratic Gaussian' (LQG) problem via the separation principle.
> Finally, many of the key objectives of study in our work are those that frequently arise in the control of linear dynamical systems -- e.g. observability and controllability Gramians, and solutions to Lyapunov and Riccati equations. We believe the relevant technical results in our paper can be of independent interest, and useful for the study of, policy search for partially observable control problems.
>
> Despite these considerations, we agree that the introduction of our initial manuscript placed too much emphasis on control of dynamical systems, which negatively impacted the framing of our problem and contribution.
> We have reworked some of this introductory material to place more emphasis on estimation and prediction, while preserving the motivation of (eventually) developing algorithms for model-free control.
> In particular,
> we have focused on motivating the study of OE based on (a) its intrinsic importance as a problem in its own right and (b) how the structure of the OE landscape is a useful didactic example to study the structure of control problems with partial observation.
> We have also sought to address your questions by making clear that the actions/decisions (i.e. the output of the policy) do not influence the state evolution of the system.
>
> Regarding the  other concern (structure and clarity of the presentation) we have include more signposting and explanation in  Section 4 to showcase the higher level logic and intuition behind the analysis, and to tie it back to the developments in the ``our techniques'' section, which was intended to introduce the key ideas underpinning our approach.
> We hope that these changes improve the clarity of exposition, but will be happy to make further modifications if there are specific aspects that remain unclear.

---

> > ### Comment · Reviewer_zdF9 · 2022-08-04
> > **Thank you for your response**
> >
> > Thank you for your response and for revising your manuscript.

---

### Official Review · Reviewer_aK2U · 2022-07-09

**Rating:** 7
**Confidence:** 2
**Soundness:** 3 good
**Presentation:** 3 good
**Contribution:** 3 good

**Summary:**

This paper proposes a direct policy search algorithm for the output estimation (OE) problem. The authors provide the global optimality guarantee of their algorithm in linear dynamical systems.

**Questions:**

See the cons above.

**Limitations:**

Yes.

**Strengths And Weaknesses:**

pros: 1. The paper discusses the problem and settings in detail.

2. The intuition behind the proposed framework is also clear and inspiring.

3. The theoretical results might be an important milestone for the community.

cons: 1. I'm not an expert in output estimation (OE) problems. Is it restricted to partially observed *linear* dynamical systems? It might be better to include the linear setting in the title.

2. Can the authors explain when suboptimal stationary points occur when the system is observable or controllable?

---

> ### Author Response · Authors · 2022-08-01
> **Initial response to Reviewer aK2U**
>
> Thank you for your feedback and questions.
> Though we do specify in the abstract that the results pertain to the "linear" output estimation problem, we agree that amending the title would be desirable. It does not seem as if we can modify the title during the rebuttal phase, but we will ask the program committee chairs if we can add "linear" to our title, as per your suggestion.
>
> We have modified Section 3.1 to make clear that suboptimal stationary points may occur at policies that are controllable (but not observable) as well policies that are observable (but not controllable).
> The revised manuscript also includes an explicit example of the latter in Appendix F.3, cf Equation F.3.
> We have also expanded Appendix F.3 to include intuitive explanations of the intuition behind the construction of these suboptimal stationary points.
> To restate here, for the case of the controllable stationary point,
> the idea is that one needs to keep track of both coordinates of the state vector to perform optimal output estimation (here, the output $\mathbf{z}$ is the system state $\mathbf{x}$, with $\mathbf{y}$ being a noisy observation thereof.) However, one can achieve controllability by only keeping track of the first coordinate, but designing the $\mathbf{A}_{\mathrm{bad}}$ such that the first coordinate excites both coordinates of the policy state (i.e. controllability). However, when it comes to ``predicting the nominal system state'', you have no information about the second coordinate.
>
> Why is this a stationary point? Here one shows that, up to first order, any perturbation of any single policy parameter either increases the cost or leaves it unchanged. The first key insight is that the controller is designed so that the estimate of the first coordinate of the system state is optimal. Thus, any perturbation which changes the prediction of the first state coordinate leads to worse performance. The second insight is that the only way to improve the performance on the second coordinate is to perturb both $\mathbf{B}_{\mathrm{bad}}$ (to measure a nonzero second coordinate of the observations) and $\mathbf{C}_\mathrm{bad}$ (to use it in the prediction) simultaneously. But this is a second order perturbation, and hence not precluded by being just a first-order stationary point.
>
> Again, we have attempted to better explain this intuition in the revised manuscript, cf. Appendix F3.

---

### Official Review · Reviewer_1P3M · 2022-07-11

**Rating:** 7
**Confidence:** 2
**Soundness:** 3 good
**Presentation:** 3 good
**Contribution:** 3 good

**Summary:**

The submission concerns a model-free policy gradient method to find the feedback gain of an observer in the linear Gaussian setting, i.e. find the Kalman gain. This optimization problem runs into issues due to the observer maintaining an internal latent Gaussian state, therefore the authors propose an additional regularization term that ensures the internal covariance is sufficient rank.

**Questions:**

Are the terms 'static' and 'dynamic' established? Because I think 'with memory' and 'memoryless' is perhaps a clearer wording for the distinction used in the paper, since it makes me think of fixed vs time-varying policies.

**Limitations:**

No issues

**Strengths And Weaknesses:**

This is obviously work of high quality. I am not well-versed in this aspect of the theory, but I can appreciate that this analysis will be of value for researchers looking at similar partially-observed problems.

However, I take some issue with the 60 pages of dense appendix. I wonder why work of such maturity and breadth is submitted to a conference and not a journal, where the review process affords more time and care. I unfortunately did not have time to review this paper in great enough detail that I think it requires, and I also think expecting a reviewer to consider ~70 pages is a rather big ask.

I read through the main text and did not have any concerns, however I found the motivation for 'model-free Kalman filtering' (29 --38), beyond the need to develop sufficient theoretical tools for further study, a bit daft. I think the theoretical study and open nature to the problem is sufficent motivation.

One aspect I found quite strange (47 -- 59) is that we typcally use a time-varying Kalman gain in state estimation / LQG, whereas this work purely looks at a fixed feedback gain in the stationary case. A discussion about the challenges of model-free methods and their analysis for the more common finite-horizon case could be beneficial to the reader.

---

> ### Author Response · Authors · 2022-08-01
> **Initial response to Reviewer 1P3M**
>
> Thank you for your feedback, and we appreciate that you did take the time to read our lengthy submission. While we agree that a detailed check is challenging given the confines of the conference review process, we also note that papers of this length have been accepted as part of the NeurIPS program, and we submitted to the conference because we think the material would be of interest to the broader community.
>
> Regarding the motivation for model-free Kalman filtering in the introduction, we have attempted to tone this down a bit in the revised manuscript; this is in keeping with an effort to place more emphasis on the importance of the problem in its own right, rather than the connections to model-free control more broadly.
>
> We agree that understanding the time-varying setting would be of interest. However, much prior work on model-free policy search in linear control also considers the "infinite-horizon" setting, in which the stationary policy parameters are known to be optimal. The reason for this is that the policy parameters can be succinctly  represented by "fixed" matrices, rather than infinite matrix recursions (in  discrete time) or matrix ODEs (in continuous time). Hence, in keeping with prior work, we elected to study the landscape of the infinite horizon loss.
>
> Regarding the nomenclature, it is our understanding that "static" and "dynamic" is the more common terminology in the control theory community.  However, your suggestion is a good one, and have happily clarified that by "static" we mean "memoryless", and by "dynamic" we mean "with memory", in the introduction.
>
> Thank you very much again for your  helpful comments. Hope our responses have addressed all of your concerns.

---

> > ### Comment · Reviewer_1P3M · 2022-08-09
> > **Post-rebuttal**
> >
> > Thanks for your comments, I am satisfied.

---

### Official Review · Reviewer_6hcE · 2022-07-12

**Rating:** 7
**Confidence:** 3
**Soundness:** 3 good
**Presentation:** 3 good
**Contribution:** 4 excellent

**Summary:**

This paper studies a policy search algorithm for the output estimation problem of a linear dynamical system, with provable convergence to the globally optimal dynamic ﬁlter.

**Questions:**

How does the regularization parameter $\lambda$ influence the optimization landscape? What happens to the limiting cases where $\lambda$ vanishes? Is there any trade-off/criteria in choosing $\lambda$ from either a theoretical or practical perspective?


**Limitations:**

I don't see any limitations or potential negative societal impact of this work.

**Strengths And Weaknesses:**

## Strengths
- This paper deals with a challenging problem of policy search in linear dynamical systems, yet provable convergence is achieved and the proof seems to be theoretically sound.
- The proposed notion of informativity characterizes the existence of critical points and hence well motivates the proposed IR-PG algorithm which appears to be novel.
- The technical tools used might be of independent interest.

## Weaknesses
- The writing and organization of this paper could be improved. Currently, it's relatively hard to follow, especially Section 4 where technical definitions and propositions are presented. It is suggested that the author provide more intuition/motivation for the analysis.

---

> ### Author Response · Authors · 2022-08-01
> **Initial response to Reviewer 6hcE**
>
> Thank you for your feedback and suggestions; in particular, we appreciate your suggestion to include more intuition and motivation for the analysis. We hoped that the ``our techniques'' discussion would explain the key challenges and our strategies for their resolution. However, we recognize that this section may have felt somewhat disconnected from Section 4. Hence, we have removed the conclusion of our submission and instead included more explanation on the  setup and structure of our analysis at the beginning of Section 4.
>
> Thank you for your question concerning the selection of the regularization parameter $\lambda$. Although the original manuscript contained some discussion of the effect of $\lambda$, it was somewhat buried in the appendices, and not highlighted in the main body. In the revised manuscript, we have placed a reference in Section 3 to Appendix F.6 where the tradeoffs involved in selecting $\lambda$ are discussed. In brief, too small $\lambda$ means that the effect of the informativity regularizer is weakened, so that $\text{trace}[\mathbf{Z}_{\mathsf{K}}^{-1}]$ may be larger. This may both increase the upper bound on the smoothness of the sublevel sets, as well as degrade the weak-PL constant. On the other hand, if $\lambda$ is very large, then the norm of the gradients grow. As Theorem 2 shows, any constant $\lambda$ suffices for our analysis.

---

> > ### Comment · Reviewer_6hcE · 2022-08-09
> > **Thanks for the replies**
> >
> > I appreciate the revision and responses the authors' made. They solve my concerns.

---

### Official Review · Reviewer_MNwe · 2022-07-22

**Rating:** 6
**Confidence:** 3
**Soundness:** 3 good
**Presentation:** 2 fair
**Contribution:** 3 good

**Summary:**

This paper introduces the first direct policy search algorithm (a model-free gradient descent type algorithm) with the global optimum convergence guarantee for solving the output estimation (OE) problem of a linear dynamical system, given Gaussian noises and partial observations.

To learn the global optimality condition related to the first-order stationary point, previous literature limited to the study of minimality condition, where gradient descent might still fail to converge to the global optimum and only converge to a suboptimal stationary point. This paper introduces a stronger condition, informativity, that is sufficient to imply that all informative stationary points are global optimum.

Thus, in order to maintain the informativity condition, the authors add regularizations to the loss function $\mathcal{L}_{\texttt{OE}}$ and show that the suboptimality of the regularized loss function $\mathcal{L}_\lambda$ upper bounds the suboptimality of the original one. The informativity is crucial to establish the weak Polyak-Łojasiewicz (PL) condition of the function $\mathcal{L}_\lambda$. Further, by adding a reconditioning step, the loss function $\mathcal{L}_\lambda$ maintains the smoothness property in the compact set. The resulted algorithm is called informative-regularized policy gradient (IR-PG). Combined with the weak PL condition and the smoothness, the authors establish $\mathcal{O}(1/T)$ sublinear global optimal convergence rate of the algorithm IR-PG.

**Questions:**

1. Is the informativity condition enough to show the weak PL condition of $\mathcal{L}_\lambda$ ? If this is the case, the authors should highlight this aspect earlier when introducing the informativity. This is an important motivation, as weak PL condition implies the global optimality of the first-order stationary point.

2. In LQR, [1] only shows that the loss function is "almost" smooth, which does not satisfy the smoothness condition. Could you bring more insights why $\mathcal{L}_\lambda$ is smooth.

3. What is the value of $d_z$ in Proposition 4.2 ? It should be provided in the proposition as well.

[1] Maryam Fazel, Rong Ge, Sham Kakade, and Mehran Mesbahi. Global convergence of policy gra- dient methods for the linear quadratic regulator. In Jennifer Dy and Andreas Krause, editors, Proceedings of the 35th International Conference on Machine Learning, volume 80 of Proceed- ings of Machine Learning Research, pages 1467–1476. PMLR, 10–15 Jul 2018.

Typos:

Line 178-179: subsampled in discrete intervals

Line 259: see Appendix E.4 for proof

Algorithm 1 Line 2: $t = 0, 1, 2$; Line 4: $\nabla_t$

Line 302: $x : f(x) \neq \infty$

Line 303: on a domain $\mathcal{K}$

Line 319: in two aspects



**Limitations:**

The authors have adequately addressed the limitations of their work.

**Strengths And Weaknesses:**

My knowledge on optimal control theory is limited. As far as I can tell, the work tackles a rather important open problem and the authors are the first to do it successfully. Most of the work in the paper neatly builds up to the main global optimum convergence result Theorem 2 in Section 3.3 and this is done in a sound manner. The discussion between the previous work minimality criterion and the informativity in Section 3.1 and the experiments in the appendix are also interesting and help to understand under which condition gradient descent of such problem will converge.

However, I found the paper to be quite hard to follow. For instance, the controllability condition in Assumption 2.4 is not defined before, which is only provided late in Appendix E. One stylistic choice by the authors that has made my reading harder is the lack of the clear definitions for many variables, e.g. $\Sigma_{11}, \Sigma_{22}, \Sigma_{11,\text{sys}}$. What is the similarity or difference between $\Sigma_{11}$ and $\Sigma_{11,\text{sys}}$ or $\Sigma_{22}$ and $\Sigma_{22,K}$ ? For instance, they can be properly defined together with the notation section in the appendix.

---

> ### Author Response · Authors · 2022-08-01
> **Initial response to Reviewer MNwe**
>
> Thank you for your feedback and suggestions. We note that the matrices in question are defined in the preliminaries, Section 2 Part A, in the Appendix. However, we understand that the paper has a considerable notational burden given the challenge in the proofs. As a consequence, we have added a notational review section to the Appendix in the revision to better aid the reader. We have also included a definition of controllability in Assumption 2.4 in the revised manuscript.
>
>
> Regarding your further concerns:
>
> 1) informativity does suffice to establish weak-PL of $\mathsf{K}$ at $\mathcal{L}_{\lambda}$; however, this is rather non-trivial and (to the best of our knowledge) requires the entire strength of our analysis framework (through DCLs). Because of this, we state in Theorem 1 (lines 233-234) that $\mathsf{K}$ satisfies the qualitative property we desire from weak PL: namely that if $\mathsf{K}$ is a stationary point and satisfies informativity, it is globally optimal.
>
> 2) The LQR objective in the cited reference [1] does also satisfy smoothness in our sense: namely, that the magnitude of their Hessian is uniformly bounded over any ``sublevel sets'' of the LQR objective. However, because their landscape analysis proceeds through a more typical PL argument with $\log(1/\epsilon)$ convergence, they required a precise upper bound on the suboptimality in a neighborhood of the optimal solution. Rather than upper bounding the Hessian as we do, they find it more convenient to work directly with the almost-smoothness property (Lemma 6 in [1]) to control a Taylor-like expansion of their loss. But this is purely a matter of convenience of analysis.
>
> 3) Here, $d_z$ is an arbitrary dimension specifying the argument of $f_{cvx}$. We have clarified this in the revision.
>
> We have also included all typo fixes, except we left "respects'" on 319 as we believe this is a valid idiom (e.g., the phrase "many respects").
>
> Thank you very much again for your careful reading and helpful comments. Hope our responses have addressed all of your concerns.

---

> > ### Comment · Reviewer_MNwe · 2022-08-08
> > **Thank you for your response**
> >
> > Thank you for your response, especially the clarification on the weak-PL condition and the smoothness property.
> >
> > Regarding to the weak-PL, I agree that it is non-trivial to prove it through the informativity condition. As I mentioned in my review, it would be nice to highlight or refer to this aspect earlier in the paper when introducing the informativity to help better understanding the connection between such condition and the global optimum convergence. Right now this property is only first presented in Page 8 which seems late to me.

---

> > > ### Author Response · Authors · 2022-08-09
> > > **Highlighting the connection between "informativity" and gradient dominance**
> > >
> > > Thank you very much for this feedback. We are in complete agreement; it would be much better to introduce the connection between informativity and gradient dominance earlier in the paper. In the revised manuscript (cf. the last sentence of "Our techniques" in Section 1.1), when we talk about quantitative notions of informativity, we have now made clear that our "differentiable convex liftings" framework allows us to establish a gradient dominance property that we term "weak-PL", in order to prove quantitative global convergence guarantees.

---

### Author Response · Authors · 2022-08-01
**Summary of revisions after initial review**

We thank all the reviewers for their comments and suggestions. We have incorporated all suggestions and typo corrections into an edited manuscript which we have uploaded. Major changes include
(a) reworking the introductory material to place more emphasis on the output estimation as a prediction problem, rather than a control problem per se,
(b) removing the conclusion to free up more text at the beginning of Section 4 to clarify the analysis and elucidate the key ideas/intuition,
(c) clarification of the existence of observable suboptimal stationary points in Section 3.1, as well as more extensive discussion of the examples of suboptimal stationary points, including an example of a policy that is observable but not controllable, cf Appendix F.3,
(d) a series of notation tables in Appendix A.3.

---

### Meta-Review · Area_Chair_B3rt · 2022-08-26

**Recommendation:** Accept
**Confidence:** Certain

**Metareview:**

The paper establishes the first global optimum convergence guarantee for solving the output estimation (OE) problem of a linear dynamical system through a model-free gradient descent algorithm. The contribution is novel and of interest for the community. All the reviewers are convinced by authors' answers and agree that the paper is a solid contribution.

**Award:**

No

---

### Decision · Program_Chairs · 2022-09-14

Accept